# Analysis of Phenolic Compounds in Commercial *Cannabis sativa* L. Inflorescences Using UHPLC-Q-Orbitrap HRMS

**DOI:** 10.3390/molecules25030631

**Published:** 2020-01-31

**Authors:** Luana Izzo, Luigi Castaldo, Alfonso Narváez, Giulia Graziani, Anna Gaspari, Yelko Rodríguez-Carrasco, Alberto Ritieni

**Affiliations:** 1Department of Pharmacy, Faculty of Pharmacy, University of Naples “Federico II,” Via Domenico Montesano 49, 80131 Naples, Italy; alfonso.narvaezsimon@unina.it (A.N.); giulia.graziani@unina.it (G.G.); annagaspari@virgilio.it (A.G.); alberto.ritieni@unina.it (A.R.); 2Department of Clinical Medicine and Surgery, University of Naples “Federico II”, Via S. Pansini 5, 80131 Naples, Italy; luigi.castaldo2@unina.it; 3Laboratory of Food Chemistry and Toxicology, Faculty of Pharmacy, University of Valencia, Av. Vicent Andrés Estellés s/n, 46100 Burjassot, Spain; yelko.rodriguez@uv.es

**Keywords:** *Cannabis sativa* L., polyphenols, UHPLC-Q-Orbitrap HRMS

## Abstract

Industrial hemp (*Cannabis sativa* L. Family Cannabaceae) contains a vast number of bioactive relevant compounds, namely polyphenols including flavonoids, phenolic acids, phenol amides, and lignanamides, well known for their therapeutic properties. Nowadays, many polyphenols-containing products made of herbal extracts are marketed, claiming to exert health-promoting effects. In this context, industrial hemp inflorescence may represent an innovative source of bioactive compounds to be used in nutraceutical formulations. The aim of this work was to provide a comprehensive analysis of the polyphenolic fraction contained in polar extracts of four different commercial cultivars (Kompoti, Tiborszallasi, Antal, and Carmagnola Cs) of hemp inflorescences through spectrophotometric (TPC, DPPH tests) and spectrometry measurement (UHPLC-Q-Orbitrap HRMS). Results highlighted a high content of cannflavin A and B in inflorescence analyzed samples, which appear to be cannabis-specific, with a mean value of 61.8 and 84.5 mg/kg, meaning a ten-to-hundred times increase compared to other parts of the plant. Among flavonols, quercetin-3-glucoside reached up to 285.9 mg/kg in the Carmagnola CS cultivar. Catechin and epicatechin were the most representative flavanols, with a mean concentration of 53.3 and 66.2 mg/kg, respectively, for all cultivars. Total polyphenolic content in inflorescence samples was quantified in the range of 10.51 to 52.58 mg GAE/g and free radical-scavenging included in the range from 27.5 to 77.6 mmol trolox/kg. Therefore, *C. sativa* inflorescence could be considered as a potential novel source of polyphenols intended for nutraceutical formulations.

## 1. Introduction

*Cannabis sativa* is an annual herbaceous plant of the Cannabaceae family native to Central Asia, but with a wide distribution over different geographical areas facilitated by climate adaptation. This plant has long been cultivated due to its large variety of applications, from textile uses to food and feed [1].

Industrial hemp, characterized by a low content of psychoactive cannabinoids, contains bioactive compounds that are known to have a wide range of important biological properties [2]. Polyphenols represent one of the most relevant compounds found in *C. sativa,* such as prenylated flavonoids, phenol amides, and lignanamides, which are specific metabolites of this plant. They are known to play multifunctional roles in the defense mechanisms of the plant, especially through their activity as antioxidants, preventing the generation of reactive oxygen species (ROS) [3,4,5,6]. In humans, polyphenols can display health-promoting effects based on the modulation of several enzymes, such us lipoxygenase and cytochrome P450 system, showing cardio or chemoprotective activity, among others [5,7]. 

For this reason, polyphenols-containing products have been marketed over the last years as food supplements and nutraceuticals, and, currently, a great variety of supplements claiming to enhance specific physiological functions are commercially available. Nutraceuticals consist of naturally-occurring active substances, which are concentrated and administered in the suitable pharmaceutical form to properly develop its pharmacological effect. Furthermore, when compared to traditional drugs, nutraceuticals appear to be generally safer, with higher bioavailability and fewer side effects [8]. The manufacturing of nutraceuticals requires isolated ingredients that have to be extracted and purified for latter uses. Since certain polyphenols naturally occur inside insoluble structures, such as vacuoles, obtention of pure compounds can become a complex process [9]. In addition, several studies reported a decrease in the bioavailability and bioaccessibility of pure polyphenols in comparison with the administration of plant extracts rich in polyphenols, which may be due to the existence of other active compounds which can establish synergistic functions with them [10,11,12]. Because of this, food supplements could be a valuable resource to consume polyphenols-containing products. They consist of extracts from herbals and botanicals than can be delivered as the same pharmaceutical forms as nutraceuticals. Some of the most prevalent plants used as a source of polyphenols are tea, coffee, apple, basil, and turmeric, among others, each one intended for specific polyphenols [13,14,15]. 

Regarding *C. sativa*, recent studies have reported the high antioxidant potential of the plant, also characterizing the major polyphenols, *N-trans*-caffeoyltyramine, and cannabisin A, B and C, and concluding that *C. sativa* would be a suitable source of polyphenols for nutraceutical or supplementation purposes [3,4,16,17,18]. Nevertheless, the most studied organs of the plant are seeds, leaves, and sprouts, whereas there is still scarce literature regarding polyphenols in inflorescences. The polyphenolic profile of *C. sativa* is variable among the different parts of the plant, and since flowers represent an important reproductive organ, high levels of colored polyphenols are expected [19].

Analysis of polyphenols in *C. sativa* samples have been previously performed using Fourier transform infrared (FTIR) spectroscopy with attenuated total reflectance (ATR) [4], mass spectrometry (MS) coupled to both high-performance liquid chromatography (HPLC), and gas chromatography (GC) [18]. High-resolution mass spectrometers, such as Orbitrap, have also been used coupled to ultra-high performance liquid chromatography (UHPLC) for the determination of polyphenols in vegetal matrices intended for nutraceutical purposes, including green tea and coffee [17,20,21,22,23]. This methodology offers higher sensitivity and specificity, allowing a precise quantification based on exact mass measurement. Therefore, the aim of this study was to (i) evaluate the antioxidant activity and total polyphenol content in different chemotypes of commercial *C. sativa* inflorescences using in vitro assays and (ii) to establish the polyphenolic profile through ultra-high-performance liquid chromatography coupled to a high-resolution Orbitrap mass spectrometry, to promote the use of this innovative source of bioactive compounds to be used in nutraceutical gformulations or for their health-promoting properties.

## 2. Results and Discussion

### 2.1. Identification of Polyphenols Compounds in C. sativa Inflorescences though UHPLC-Q-Orbitrap HRMS

Identification of individual phenolic acids and flavonoids was conducted through UHPLC-Q-Orbitrap HRMS. By a combination of MS and MS/MS spectra, a total of 22 different polyphenolic compounds were identified from different samples of *C. sativa* inflorescences (Appendix A). Table 1 shows all mass parameters including adduct ion, theoretical and measured mass (*m/z*), accuracy and sensitivity.

Experiments were achieved in ESI^−^ mode. All of the studied analytes exhibited better fragmentation patterns producing the quasi-molecular ion [M − H]^−^. After full scan analysis, the accurate mass of the characteristic ions (precursor ions) was included in an inclusion list.

Full-scan HRMS data acquisition captures all sample data, enabling the identification of untargeted compounds and retrospective data analysis without the need to re-run samples. The confirmation of the structural characterization of untargeted analytes was based on the accurate mass measurement, elemental composition assignment, and MS/MS spectrum interpretation (Appendix A). 

Optimal separation of all the investigated analytes was carried out in a total run time of 20 min. The identification of structural isomers: catechin and epicatechin (*m/z* 289.07176); luteolin and kaempferol (*m/z* 285.04046), was achieved by comparing the retention times of the peaks with those of standards (Appendix A).

Sensitivity was evaluated by the limit of detection (LOD) and limit of quantification (LOQ). The LOD was defined as the minimum concentration, where the molecular ion could be identified with a mass error below 5 ppm, and the LOQ was set as the lowest concentration of the analyte that produced a chromatographic peak with a precision and accuracy <20%.

Quantitative determination of target analytes (*n* = 16) was performed using calibration curves at eight concentration levels. Each calibration curve was prepared in triplicate. We obtained regression coefficients >0.990. Quantification of compounds (*n* = 6) that had no standard to generate a curve was based on a representative standard of the same group.

### 2.2. Quantification of Phenolic Acids and Flavonoids in C. sativa Inflorescences

#### 2.2.1. Phenolic Acids

The predominant lignanamides (cannabisin A, B, and C) and phenolic amide (*N-trans*-caffeoyltyramine) found in hemp were evaluated in the assayed samples. Lignanamides and phenolic amides belong to the lignan class of compounds, and the basic unit consists of tyramine condensed with CoA-esters of *p*-coumaric, caffeic and coniferic acid, as suggested by Flores-Sanchez [24]. Table 2 shows the results here obtained expressed as the average content and concentration range of the phenolic acids and flavonoids detected in different hemp varieties. In the here analyzed samples, lignanamides represented from 0.02% to 0.47% of total polyphenols in a concentration range between 0.10 and 2.2 mg/kg. Cannabisin A was found as the most commonly detected lignanamide ranging from 0.01 (Tiborszallasi) up to 2.86 mg/kg (Kompolti), with a mean value of 1.0 mg/kg for all cultivars. Cannabisin B was found at levels three times lower with respect to Cannabisin A, ranging from 0.4 to 0.5 mg/kg. In addition, when cannabisin A was found at very low concentrations, cannabisin B was not detected. Cannabisin C showed to be the less relevant lignanamide, quantified between 0.003 and 0.38 mg/kg. Concerning the occurrence of lignanamides in *C. sativa* seed, available studies reported the highest concentration up to thousands milligram per kilogram [25]. As far as phenol amides were concerned, *N-trans*-caffeoyltyramine was quantified at a concentration range from 0.1 (Kompolti) to 76.2 mg/kg (Carmagnola Cs), with a mean value of 23.7 mg/kg for all cultivars. These levels are in line with the data reported in hemp seed [25]. Lignanamides and phenolic amides are known to have a wide range of important biological properties, including antioxidant, anti-inflammatory, and antihyperlipidemic activities [26,27,28,29,30]. Apart from those, some important hydroxycinnamic acids (chlorogenic acid, caffeic acid, *p*-coumaric acid, and ferulic acid) were evaluated in the here analyzed inflorescences samples. This important class of phenolic acids represented from 18.6% to 29.7% of total polyphenols found in samples. Among the hydroxycinnamic acids, *p*-coumaric acid was quantified at concentrations significantly greater (*p* < 0.05) than the other related compounds in all hemp cultivars analyzed except in Kompolti samples. Moreover, the most common hydroxycinnamic acids found in Kompolti cultivar was ferulic acid at an average content of 19.7 mg/kg (range from 3.0 to 35.6 mg/kg). Caffeic acid was detected in the lowest amount for all the analyzed cultivars. Carmagnola Cs hemp variety showed the highest concentration of hydroxycinnamic acids compared with other varieties at an average content of 85.4 mg/kg. On the other hand, the observed concentration variability of phenolic acids may be a result of the influence of many biotic and abiotic factors that play an important role in the biosynthetic process of the studied compounds [31].

#### 2.2.2. Flavonoids

Flavonoids are plant-derived phytochemicals that accounted for over 80% of the phenolic components in the assayed samples. Flavonoids, namely flavones, flavanones, flavonols, and flavanols, including their main aglycones, glycosides, and methylated derivatives, have been quantified in the assayed samples as shown in Table 2. Flavones represented the highest proportion of flavonoids found in analyzed samples, ranging from 30.1% (Carmagnola Cs) to 35.4% (Kompolti) of total polyphenols, cannflavin A and cannflavin B being the most commonly detected flavones, with a mean value of 61.8 and 84.5 mg/kg, respectively. These levels found in inflorescences showed a ten-to-hundred-fold increase when compared to leaves samples previously analyzed by Pollastro et al., [5] who reported cannflavin A and B at 6 and 0.8 mg/kg, respectively. Cannflavins A and B, methylated isoprenoid flavones, appear to be *Cannabis* specific and are known to exert a potent anti-inflammatory activity [32]. In this context, Werz et al. [33] reported that cannflavin A and B were able to inhibit the production of pro-inflammatory prostanoids and leukotrienes in in vitro assays. Moreover, Barrett et al. concluded that cannflavins A and B promoted the inhibition of PGE2 in human cells up to thirty times in relation to aspirin [34,35]. In addition, cannflavin A showed a neuroprotective effect against amyloid β-mediated neurotoxicity in PC12 cells [36]. As shown in Table 2, among flavonols, quercetin-3-glucoside was quantified at concentrations significantly greater (*p* < 0.05) than the other related compounds, ranging from 2.2 (Kompolti) to 285.9 mg/kg (Carmagnola Cs) with a mean value of 78.4 mg/kg. On the other hand, flavanols mainly represented by catechin and epicatechin were detected at mean values of 53.3 and 66.2 mg/kg for all cultivars, respectively. Regarding epicatechin, Carmagnola Cs cultivar showed a two-fold increase in the mentioned-flavonol compared with other cultivars assayed. The minor flavonoids compound detected was naringenin.

Overall, the data clearly indicate that Carmagnola Cs cultivar showed the highest concentrations of the investigated polyphenols (743.5 mg/kg) compared to the other cultivars. 

Industrial hemp inflorescence may represent an innovative source of bioactive compounds, such as a high content of cannflavin A and B, which appear to be cannabis-specific to be used in nutraceutical formulations.

### 2.3. Total Phenolic Contents and Antioxidant Activity of C. sativa Extracts

#### 2.3.1. Total Phenolic Contents of *C. sativa* Extracts

The amounts of total phenolic contents of *C. sativa* extracts were examined. The results are summarized in Table 3. Total phenolics, flavonoids, and phenolic acids ranged from 10.510 to 48.875 mg GAE/g for the different cultivars of *C. sativa*. Among the studied cultivars, the highest total phenols content resulted from Carmagnola Cs with the content of 41.517 mg GAE/g and, the lower amount emerges for the Kompolti variety (10.510 mg GAE/g). Total polyphenol content quantified in analyzed samples was similar to those recently reported by Vonapartis et al., [37] in hemp seeds samples (*n =* 10) in a concentration range from 13.68 to 51.60 mg GAE/g.

*C. sativa* is a remarkable plant widely investigated by several researchers, given its rich fount of valuable natural components. Apart from cannabinoids production, the *C. sativa* plant is also able to synthesize non-cannabinoids second metabolites possessing benefic effects for human health [1,38]. A summary of the available surveys of total polyphenols contents of the different parts of *C. sativa* is shown in Table 4. In spite of several experiments were done on this plant [3,4,17,38,39,40,41,42,43], the study of inflorescence remains low, making it difficult the comparison of results. Recently, Ferrante et al. [25] investigated the total phenolic content in the water fraction extracted from aerial flowers of Carmagnola Cs cultivar, reporting a lower value compared to the results here obtained.

With respect to other parts of the same plant, the given TPC in *C. sativa* inflorescences was significantly (*p* < 0.05) higher than the reported contents, resulting in the highest potential fount of phenols.

The correlation between TPC and UHPLC-Q-Orbitrap HRMS findings was evaluated through Pearson’s correlation coefficient (PCC). The results showed a strong positive correlation (PCC = 0.892). Scientific evidence suggests that using some combination of assays is the best approach to properly characterize the phenolic composition [44].

#### 2.3.2. Antioxidant Activity of *C. sativa* Extracts

The results of antioxidant activity evaluated through DPPH free radical-scavenging activity are tabulated in Table 3 and expressed as mmol trolox/kg. A calibration curve of inhibition, built with trolox®, was employed as a positive control of the assay. Antioxidant activity was included in the range of 27.532 to 77.578 mmol trolox/kg (average 54.401 mmol trolox/kg) for the different cultivars of *C. sativa*. The highest antioxidant capability was shown, in this case, in Carmagnola Cs (*p* < 0.05) with a content of 77.578 mmol trolox/kg and the lower amount in the Antal variety (27.532 mmol trolox/kg). 

Antioxidant capacity is largely used as a parameter to characterize bioactive components from foods or medicinal plants. The assessment of polyphenols contents in inflorescences of *C. sativa* extract is useful to define their potential antioxidant value and their free radical scavenging capacity. Increasing evidence reports that antioxidants may protect cell constituents against oxidative damage and shrink the risk of various diseases connected to oxidative stress [45]. 

The synergy effect or interaction of phenolic compounds in food products contribute, for the most part, to the overall antioxidant capacity, despite bioactivity decoupled from phenolic compounds being reported in the literature [46,47]. 

Several researchers have studied the bioactivity of various parts of *C. sativa* [3,4,17,39,40,41,42,43], and, on the whole, the extracts were found to be rich sources of bioactive compounds. As regards antioxidant capability, Smeriglio et al. [40] reported contents of 146.7 mmol trolox/100 g in seed oil from Finola cultivar and lower amounts for the lipophilic fraction (0.125 mmol trolox/100 g) and hydrophilic fractions (0.038 mmol trolox/100 g). Instead, Mikulek et al., [43] have reported an amount of 158 mmol trolox/100 kg in flour.

Currently, scarce data on the antioxidant activity of *C. sativa* inflorescence are available in the literature. Compared to other findings that reported the antioxidant activity of the different parts of *C. sativa,* the inflorescence showed the highest properties. 

## 3. Materials and Methods

### 3.1. Reagents and Materials

A total of twenty-seven samples of industrial *C. sativa* female inflorescences of different varieties, including Kompolti (*n* = 9), Tiborszallasi (*n* = 7), Antal (*n = 7*), and Carmagnola Cs (*n* = 4) were provided by several hemp farmers located in Italy. All inflorescences were harvested in October 2019 and fulfilled the requirements set at EC regulation (No 809/2014) [48] regarding their psychoactive cannabinoids content. The samples were dried at 36 °C using a forced-air laboratory oven until the sample moisture reached a level from 8% to 12%. The samples were milled into powder using a laboratory mill (particle size 200 µm) and then stored at 4 °C until analysis.

The standards of polyphenols (purity >98%) were purchased from Sigma Aldrich (Milan, Italy), and included: chlorogenic acid, caffeic acid, *p-*coumaric acid, ferulic acid (Hydroxycinnamic Acids); rutin, quercetin-3-glucoside, kaempferol-3-*O*-glucoside, quercetin, kaempferol (Flavonol); luteolin-7-*O*-glucoside, apigenin-7-*O*-glucoside, luteolin, apigenin (Flavones); catechin, epicatechin (Flavanols) and naringenin (Flavanone).

Due to the lack of analytical standards, the identification of polyphenols (*n* = 6) including cannabisin A, cannabisin B, cannabisin C (Lignanamide), cannflavin A, cannflavin B (Flavanone) and *N-trans*-caffeoyltyramine (Phenolic amides) was carried out by a post-target screening.

Methanol (MeOH), water (LC-MS grade) were acquired from Merk (Darmstadt, Germany), and formic acid (mass spectrometry grade) was purchased from Fluka (Milan, Italy).

### 3.2. Polyphenols Extraction

Extraction has been carried out according to the procedure described by Calzolari et al., [49] with some modifications. Briefly, 100 mg of sample was suspended in 15 mL of methanol, the mixture was vortexed intensively for 3 min and sonicated in the dark, at 4 °C, for 30 min. Then, the mixture was centrifuged at 5000× *g* at 4 °C for 10 min. The supernatants were pooled, filtered through 0.2 μm syringe filters (26 mm, RC membrane, Phenomenex, Castel Maggiore, Italy), and an aliquot introduced into a chromatography vial.

### 3.3. Determination of Total Phenolic Content (TPC)

Total phenolic content was performed according to the Folin–Ciocalteu method [50] with slight modifications. Briefly, 125 μL of extract sample was diluted in 500 μL of deionized water, then 125 μL of the Folin–Ciocalteu reagent was added to the mixture, followed by 6 min of incubation at room temperature. Afterward, 1.25 mL of 7.5% of sodium carbonate solution and 1 mL of deionized water were added in the mixture. The absorbance at 760 nm after 90 min of incubation in the dark was measured. The TPC of inflorescence samples was expressed as mg of gallic acid equivalents (GAE)/g of sample.

### 3.4. Determination of 1,1-Diphenyl-2-picrylhydrazyl (DPPH) Free Radical-Scavenging

The radical-scavenging activity of the sample extract was determined using the method suggested by Brand-Williams et al. [51] with some modifications. Briefly, to obtain the DPPH radical working solution, the DPPH standard (4 mg in 10 mL) was diluted with methanol until the absorbance value reached 0.90 (±0.02) at 517 nm. Then, 200 μL of sample extract was added to 1 mL of DPPH radical working solution. The mixture was shaken vigorously, and then the decrease absorbance after 10 min at 517 nm was measured. The results were expressed as mmol trolox equivalents (TE)/kg of the sample.

### 3.5. Ultra-High Performance Liquid Chromatography and Orbitrap High-Resolution Mass Spectrometry Analysis

The polyphenolic profile was analyzed by Ultra High-Pressure Liquid Chromatograph (UHPLC, Dionex UltiMate 3000, Thermo Fisher Scientific, Waltham, MA, USA) equipped with a degassing system, a Quaternary UHPLC pump working at 1250 bar, and an autosampler device. Chromatographic separation of polyphenols was performed with a thermostated (T = 25 °C) Kinetex 2.6 µm Biphenyl (100 × 2.1 mm, Phenomenex) column. The injection volume was 2 μL. The mobile phase consisted of a binary solution: water (phase A) and methanol (phase B), both mobile phases contained 0.1% of formic acid. A gradient elution program was applied as follows: an initial 5% B, increased to 30% B in 1.3 min, and a new to 100% B in 8 min. The gradient was held for 2 min at 100% B and reduced to 5% B in 2 min. The flow rate of 0.2 mL/min. Afterward, the gradient switched back to 5% in 2 min, and another 2 min for column re-equilibration at 5%. The UHPLC system was coupled to a Q-Exactive Orbitrap mass spectrometer (UHPLC, Thermo Fischer Scientific, Waltham, MA, USA). An ESI source (Thermo Fisher Scientific, Waltham, MA, USA) was operated in negative ion mode (ESI-) setting two scan events (Full ion MS and All ion fragmentation, AIF) for all compounds of interest. Full scan data were acquired at a resolving power of 35,000 FWHM (full width at half maximum). Ion source parameters were spray voltage 2.8 kV (negative mode), capillary temperature 310 °C, S-lens RF level 50, sheath gas pressure (N_2_ > 95%) 35, auxiliary gas (N_2_ > 95%) 10, auxiliary gas heater temperature 350 °C. The value for the automatic gain control (AGC) target was set at 3 × 10^6^, a scan range of *m/z* 90 to 1000 was chosen, and the injection time was set to 200 ms. The scan-rate was set at 2 scans/s. Data analysis and processing were performed using Xcalibur software, v. 3.1.66.10. 

For the scan event of AIF, the parameters were set as follows: mass resolving power of 17,500 FWHM at 200 ms; scan time = 0.10 s. The collision energy was varied in the range of 10 to 45 eV to obtain representative product ion spectra. Data processing was performed by the Quan/Qual Browser Xcalibur software, v. 3.1.66.10 (Xcalibur, Thermo Fisher Scientific, Waltham, MA, USA). Detection was based on calculated exact mass with a mass error below 5 ppm and on the retention time of the molecular ion; while regarding the fragments on the intensity threshold of 1000 and a mass tolerance of 5 ppm. Quantitative results were obtained working in scan mode with HRMS exploiting the high selectivity achieved in full-scan mode, whereas MS/HRMS information was used for confirmatory purposes.

### 3.6. Statistics and Data Analysis

Values were expressed as the average values and concentration range of triplicate measurements. The differences between average values were evaluated by using Tukey’s test at the level of significance *p* < 0.05. Statistical analysis was performed using STATA 12 (STATA corp LP, College Station, TX, USA).

## 4. Conclusions

Even if studies regarding the beneficial effects of hemp seeds, oils, and leaves are numerous, and research on *C. sativa* extract is constantly in progress, there are few references concerning the biological activities and the potential health benefits of *C. sativa* inflorescence. A comprehensive analysis of the bioactivity for Kompolti, Tiborszallasi, Antal, and Carmagnola Cs cultivar of *C. sativa* inflorescences and polyphenols characterization through UHPLC-Q-Orbitrap spectrometry measurement were carried out in this research work for the first time.

A comparison of the studied cultivars showed that Carmagnola CS had the highest investigated polyphenols amount (sum average of 743.5 mg/kg), TPC content (33.2 ± 0.5 mg GAE/g) as well as free radical-scavenging activity (63.6 ± 0.9 mmol trolox/kg), thus, appeared to be the most promising cultivar. In spite of the renovated interest for this cultivar, data about the correlation on bioactivity and cultivar are still fragmentary. 

Our results highlighted the possibility of also using this part of the plant, which represents a valuable source of natural antioxidants and a rich fount of polyphenols, including cannflavins, which represent bioactive compounds not common in other typical plants. It is, therefore, desirable to continue to expand the understanding of this actual topic to estimate their efficacy for future applications for nutraceutical purposes.

## Figures and Tables

**Table 1 molecules-25-00631-t001:** Chromatographic and spectrometric optimized parameters including retention time, adduct ion, theoretical and measured mass (*m/z),* accuracy and sensibility for the investigated analytes *(n* = 22).

Compound	Retention Time (min)	Chemical Formula	Adduct Ion	Theoretical Mass (*m*/*z*)	Measured Mass (*m*/*z*)	Accuracy (Δ mg/kg)	LOD (mg/kg)	LOQ (mg/kg)
Catechin	7.65	C_15_H_14_O_6_	[M − H]^−^	289.07176	289.07224	1.6605	0.0015	0.0046
Chlorogenic acid	8.13	C_16_H_18_O_9_	[M − H]^−^	353.08780	353.08798	0.5098	0.0012	0.0036
Caffeic acid	8.24	C_9_H_8_O_4_	[M − H]^−^	179.03498	179.03455	−2.4018	0.0007	0.0020
Epicatechin	8.51	C_15_H_14_O_6_	[M − H]^−^	289.07176	289.07196	0.6919	0.0014	0.0043
Luteolin-7-*O*-glucoside	9.23	C_21_H_20_O_11_	[M − H]^−^	447.09328	447.09366	0.8499	0.0008	0.0025
*p*-Coumaric acid	9.31	C_9_H_8_O_3_	[M − H]^−^	163.04001	163.03937	−3.9254	0.0006	0.0018
Caffeoyl tyramine	9.46	C_17_H_17_NO_4_	[M − H]^−^	298.10848	298.10910	2.0798	-	-
Rutin	9.79	C_27_H_30_O_16_	[M − H]^−^	609.14611	609.14624	0.2134	0.0012	0.0035
Ferulic acid	9.88	C_10_H1_0_O_4_	[M − H]^−^	193.05063	193.05016	−2.4346	0.0018	0.0054
Quercetin-3-glucoside	9.93	C_20_H_20_O_12_	[M − H]^−^	463.08820	463.08862	0.9070	0.0017	0.0052
Kaempferol-3-*O*-glucoside	10.36	C_21_H_20_O_11_	[M − H]^−^	447.09323	447.09360	0.8276	0.0008	0.0025
Apigenin-7-glucoside	10.36	C_21_H_20_O_10_	[M − H]^−^	431.09837	431.09836	−0.0232	0.0004	0.0013
Cannabisin A	10.54	C_34_H_30_N_2_O_8_	[M − H]^−^	593.19294	593.19281	−0.2192	-	-
Quercetin	11.00	C_15_H_10_O_7_	[M − H]^−^	301.03538	301.03508	−0.9966	0.0021	0.0064
Luteolin	11.25	C_15_H_10_O_6_	[M − H]^−^	285.04046	285.04050	0.1403	0.0004	0.0012
Cannabisin B	11.41	C_34_H_32_N_2_O_8_	[M − H]^−^	595.20859	595.20709	−2.5201	-	-
Kaempferol	11.60	C_15_H_10_O_6_	[M − H]^−^	285.04046	285.04086	1.4033	0.0005	0.0014
Naringenin	11.78	C_15_H_12_O_5_	[M − H]^−^	271.06120	271.06146	0.9592	0.0005	0.0015
Apigenin	11.85	C_15_H_10_O_5_	[M − H]^−^	269.04555	269.04572	0.6319	0.0004	0.0011
Cannabisin C	12.34	C_35_H_34_N_2_O_8_	[M − H]^−^	609.22424	609.22485	1.0013	-	-
Cannflavin B	13.77	C_21_H_20_O_6_	[M − H]^−^	367.11871	367.11871	0.000	-	-
Cannflavin A	14.84	C_26_H_28_O_6_	[M − H]^−^	435.18131	435.18143	0.2757	-	-

**Table 2 molecules-25-00631-t002:** Polyphenols content in the analyzed *Cannabis sativa* samples (*n* = 22). Results are shown based on the different cultivar *C. sativa* inflorescences.

Sample	Kompolti (*n* = 9)		Tiborszallasi (*n* = 7)		Antal (*n* = 7)		Carmagnola Cs (*n* = 4)	
	Average (mg/kg)	Range (mg/kg)	Average (mg/kg)	Range (mg/kg)	Average (mg/kg)	Range (mg/kg)	Average (mg/kg)	Range (mg/kg)
Phenolic acids								
Hydroxycinnamic acids								
Chlorogenic acid	12.0	2.2–28.2	9.0	2.0–20.5	10.1	3.5–23.6	15.0	11.1–22.1
Caffeic acid	1.4	0.4–2.8	4.3	1.2–6.4	3.3	1.6–5.6	3.9	2.9–4.6
*p*-Coumaric acid	13.1	0.5–28.0	37.3	15.5–84.7	28.2	5.1–105.8	41.1	18.1–93.0
Ferulic acid	19.7	3.0–35.6	26.0	14.7–35.3	18.9	4.7–30.6	25.5	20.2–33.4
SUM	46.2		76.5		60.5		85.4	
Lignanamides								
Cannabisin A	1.1	0.1–2.9	0.01	0.005–0.01	1.5	1.2–1.8	1.6	0.09–2.85
Cannabisin B	0.40	0.02–1.1	-	-	0.6	0.4–0.7	0.5	0.02–1.15
Cannabisin C	0.10	0.01–0.35	0.09	0.01–0.27	0.14	0.01–0.38	0.02	0.003–0.05
SUM	1.60		0.1		1.7		2.12	
Phenolic amides								
*N-trans*-Caffeoyltyramine	17.6	0.1–59.2	15.3	4.7–30.6	25.8	5.7–44.9	36.1	5.3–76.2
FLAVONOIDS								
Flavonol								
Rutin	13.8	1.7–42.8	27.6	14.1–38.6	12.5	2.4–24.1	27.2	7.8–60.8
Quercetin-3-glucoside	37.6	2.2–87.1	94.3	23.2–269.8	55.8	10.0–172.1	126.1	58.9–285.9
Kaempferol-3-*O*-glucoside	6.5	0.1–15.3	15.4	4.2–44.5	9.9	1.6–29.6	26.6	12.6–46.6
Quercetin	12.1	6.2–24.6	15.8	6.3–26.0	10.3	6.5–16.8	28.8	8.2–58.5
Kaempferol	3.4	0.7–7.6	5.1	1.3–8.7	2.4	0.3–3.9	9.2	0.6–13.6
SUM	73.4		158.2		90.9		217.8	
Flavones								
Cannflavin A	55.4	35.2–130.0	72.9	30.9–107.5	51.3	19.6–106.5	67.7	28.4–118.6
Cannflavin B	55.4	30.2–110.6	86.7	26.2–215.5	98.8	32.9–182.5	97.2	11.9–154.4
Luteolin-7-*O*-glucoside	9.4	3.3–19.0	13.4	2.1–42.8	13.9	8.8–17.7	22.2	1.1–52.6
Apigenin-7-*O*-glucoside	1.9	0.1–5.6	2.5	0.1–6.7	1.1	0.2–3.0	2.6	0.8–6.5
Luteolin	12.2	0.8–23.5	19.0	6.3–38.2	8.6	0.9–14.3	25.7	16.9–38.2
Apigenin	5.4	2.1–11.7	6.5	2.4–16.3	4.1	0.4–8.5	8.7	6.3–13.9
SUM	139.7		201.0		27.7		59.2	
Flavanols								
Catechin	85.3	0.1–334.0	16.8	2.6–43.4	40.9	1.9–155.2	70.1	39.2–115.2
Epicatechin	30.7	1.6–88.4	64.2	10.6–183.7	63.2	11.1–156.4	106.9	21.5–194.6
SUM	116.0		81.0		104.1		177.0	
Flavanone								
Naringenin	0.50	0.01–1.10	0.5	0.3–1.0	0.9	0.5–2.0	1.0	0.7–1.8
Total bioactive	377.9		532.6		461.7		743.5	

**Table 3 molecules-25-00631-t003:** Total polyphenol compounds expressed as mg GAE/g and antioxidant activity expressed as mmol trolox/kg of *C. sativa* extracts.

Cultivar	TPC		DPPH	
	Average (mg GAE/g)	Range (mg GAE/g)	Average (mmol trolox/kg)	Range (mmol trolox/kg)
Kompolti (*n* = 9)	26.2 ± 0.5	10.5–47.2	46.7 ± 0.7	36.6–55.0
Tiborszallasi (*n* = 7)	29.9 ± 0.4	21.9–42.9	61.3 ± 0.9	50.8–72.7
Antal (*n* = 7)	30.7 ± 0.5	17.0–48.9	45.9 ± 0.4	27.5–67.5
Carmagnola Cs (*n* = 4)	33.2 ± 0.5	26.3–41.5	63.6 ± 0.9	59.1–77.6

*GAE: gallic acid equivalents; TPC: total phenolic content; DPPH: free-radical scavenging; trolox: equivalent antioxidant capacity (TEAC).

**Table 4 molecules-25-00631-t004:** Recent surveys reporting the total phenolic content (mg GAE/g) in different parts of C. *sativa* samples.

Part of Plant	TPC (mg GAE/g)	References
Leaves	0.09–0.56	[3]
Seeds	0.77–51.6	[4,17,37,39]
Oil	0.02–2.67	[4,39,40]
Flour	0.74–1.71	[4,39,43]
Sprouts	6.16	[17]
Aerial parts	5.85–17.05	[41]
Flowers	4.7–8.1	[42]
Inflorescences	10.51–52.58	Current work

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
