# Peer review of "Analysis of Phenolic Compounds in Commercial Cannabis sativa L. Inflorescences Using UHPLC-Q-Orbitrap HRMS"

_molecules, 2020, doi:10.3390/molecules25030631_

Round 1

Reviewer 1 Report

Authors reported characterization of Phenolic Compounds in Commercial Cannabis Sativa L. inflorescences. Their findings are important for future studies. 

Minor revisions:

Authors selected four chemotypes for the study. It would be good if they can give justification for the selection of four chemotypes. Authors wrote the manuscript thoroughly. However, inclusion of Figures such as the intact mass spectra of polyphenols identified will increase the impact of the manuscript (maybe as supporting information). Table2. polyphenols content: Great variation between samples within a group; for example, in Carmognola (n=4) Quercetin-3-glucoside amount range from 58.9-285.9 and the authors claimed the amount reach 285.9. Was this result reproducible?

Author Response

Point 1: Authors selected four chemotypes for the study. It would be good if they can give justification for the selection of four chemotypes. Authors wrote the manuscript thoroughly.

Response 1: Despite several varieties are available on the market, chemotypes analyzed in the current study (Kompolti, Tiborszallasi, Antal and Selected Carmagnola) appear to be the best-known hemp cultivars and mainly cultivated in Italy for commercial purposes. Therefore, the authors decided to study the cultivars above cited.

Point 2: However, inclusion of Figures such as the intact mass spectra of polyphenols identified will increase the impact of the manuscript (maybe as supporting information).

Response 2: As suggested by reviewer 1, the authors included the mass spectra of polyphenols identified in supplementary materials (Supplementary Figure S3).

Point 3: Table2. polyphenols content: Great variation between samples within a group; for example, in Carmognola (n=4) Quercetin-3-glucoside amount range from 58.9-285.9 and the authors claimed the amount reach 285.9. Was this result reproducible? 

Response 3: Reproducibility (%RSD) was calculated after analyzing samples in triplicate. All compounds showed %RSD below 20%, including quercetin-3-glucoside. The variability of results could be due to different factors i.e. genetic variants, soil type, cultivation techniques, storage and environmental conditions, which can affect the total content of active ingredients as reported by several scientific studies. Samples analyzed in the present study had different deriving and therefore, all these possible factorial differences may explain the width observed in the concentration ranges.

The authors thank reviewer 1 for evaluating the manuscript.

Reviewer 2 Report

I would  considered a different word than “Characterization” in the title; evaluation/examination/analysis? Table text of table 1 seems incomplete. I would like to see UHPLC chromatograms exemplified for the four different C. sativa cultivars, illustrating the different polyphenols found, together with a LC-chromatogram of the standards used. This point is also linked to point no. 4) below. Standardization/identification of compounds:

Example: several of the flavonoid masses have isomers, which means that the isomers will also have similar HR-MS values. For correct identification, you need similar retention-time linked to the mass you find for the standard compound, or fragmentation patterns.

I would like to see a better documentation of your qualitative analysis, on which the quantitative data depends on.

5. I would like to see some critical discussion comparing the TPC values and the qMS values.

6. It seems to me that the Journal-title of the reference seems missing several places?

Author Response

Point 1: I would considered a different word than “Characterization” in the title; evaluation/examination/analysis?

Response 1: As the reviewer 2 suggested, the authors changed the word “Characterization” to “Analysis” in the title.

Point 2: Table text of table 1 seems incomplete.

Response 2: As suggested by reviewer 2, the authors changed table 1 description as “Table 1. Chromatographic and spectrometric optimized parameters including retention time, adduct ion, theoretical and measured mass (m/z), accuracy and sensibility for the investigated analytes (n=22)”.

Point 3: I would like to see UHPLC chromatograms exemplified for the four different C. sativa cultivars, illustrating the different polyphenols found, together with a LC-chromatogram of the standards used. This point is also linked to point no. 4) below. Standardization/identification of compounds: Example: several of the flavonoid masses have isomers, which means that the isomers will also have similar HR-MS values. For correct identification, you need similar retention-time linked to the mass you find for the standard compound, or fragmentation patterns. I would like to see a better documentation of your qualitative analysis, on which the quantitative data depends on.

Response 3: As suggested by reviewer 2, total ion chromatogram (TIC) of C. sativa extract was reported in supplementary materials. Cultivars showed quantitative differences but we could not notice a relevant variance among the different total ion chromatograms. In supplementary figure S2, the authors also reported the plots of the extracted ion chromatograms (EICs) attaching the retention times of the different twenty-two analytes, including isomeric compounds luteolin-kaempferol and catechin-epicatechin.

Point 4: I would like to see some critical discussion comparing the TPC values and the qMS values.

Response 4: As suggested by reviewer 2, the authors compared the TPC and qMS values in the discussion section as: “The correlation between TPC and UHPLC-Q-Orbitrap HRMS findings was evaluated through Pearson's correlation coefficient (PCC). The results showed a positive strong correlation (PCC=0.892). Scientific evidence suggests that using some combination of assays is the best approach to properly characterize the phenolic composition [47].”

Point 5: It seems to me that the Journal-title of the reference seems missing several places?

Response 5: As rightly suggested by reviewer 2, the authors added the Journal-title in the references.

The authors thank reviewer 2 for evaluating the manuscript.

Reviewer 3 Report

Please see in the attached file

Author Response

Point 1: Generally, flavones and flavonol compounds have different antioxidant activities because of their various Log P0 values and number of phenol groups. In table 2, there are some flavones and flavonol compounds, what about their antioxidant activities? If possible, the authors please measure the antioxidant activities of some important cannflavins and their related cannflavonols by DPPH method.

Response 1: In the current study, the antioxidant activity was measured on the total polar extracts of different varieties of Cannabis sativa in which were contained different antioxidant molecules including the compounds reported in table 2. The authors consider an interesting approach the analysis of individually some important active ingredients, such as cannflavins, but it would not completely fit the main goal of this study, which was to propose C. sativa polar extract as an innovative source of bioactive compounds. Anyways, authors will consider this an idea for a future work.

Point 2: Many unsaturated natural products may undertake peroxidation and eventually convert to epoxides. Are they stable in the solution conditions under sunlight? It probably be oxidized to the epoxides and crosslinked products.

Response 2: As rightly written by reviewer 3, several studies provide evidence that unsaturated natural products are not stable in the solution conditions under sunlight. On the other hand, plant antioxidants i.e. polyphenols protect unsaturated compounds against oxidative stress, so the here-analyzed C. sativa inflorescence extracts could represent a valuable source of natural antioxidants and a rich fount of polyphenols.

Point 3: Moreover, what are the major biological functions of the specific cannflavin molecules? If possible, the authors please discuss more in the main text or cite more references.

Response 3: As suggested by reviewer 3, the authors added in the text details about major biological functions of the specific cannflavin as: “In this context, Werz et al., [33] reported that cannflavin A and B were able to inhibit the production of pro-inflammatory prostanoids and leukotrienes in in vitro assay. Moreover, Barrett et al. concluded that cannflavins A and B promoted the inhibition of PGE2 in human cells up to thirty times in relation to aspirin [34, 35]. In addition, cannflavin A showed a neuroprotective effect against amyloid β-mediated neurotoxicity in PC12 cells [36]”.

Point 4: The format of references needs to be re-organized.

“[35] …toxicological aspects, and analytical determination. Journal name?. 2019, 6, (1), 31.”;

“[43] Olszowy, M. J. P. P.; Biochemistry, What is responsible for antioxidant properties of polyphenolic compounds from plants? 2019..

”[44] Piccolella, S.; Crescente, G.; Candela, L.; Pacifico, S. J. J. o. p.; analysis, b., Nutraceutical Polyphenols: New analytical challenges and opportunities. 2019…” ,etc. The authors please carefully check the references.

Response 3: As suggested by reviewer 3, the authors checked the references.

The authors thank reviewer 3 for evaluating the manuscript.

Round 2

Reviewer 2 Report

I still feel that the quantitative determination is a bit unclear, so this part may be improved.

Author Response

Point 1: I still feel that the quantitative determination is a bit unclear, so this part may be improved.

Response 1: As rightly suggested by reviewer 2, the authors added details about the quantitative determination of analytes in the manuscript.

The authors thank the reviewer for evaluating the manuscript.